# Magnetic Resonance Imaging (MRI) Findings in COVID-19 Associated Encephalitis

**Manoj Tanwar [1,\*], Aparna Singhal [1], Mohammadreza Alizadeh [2] and Houman Sotoudeh [1]**

1    Department of Radiology, Heersink School of Medicine, University of Alabama at Birmingham, Birmingham, AL 35233, USA

2    Physiology Research Center, Iran University of Medical Sciences, Tehran 14665-354, Iran

\*    Correspondence: mtanwar@uabmc.edu

**Abstract:** We conducted this study to investigate the scope of the MRI neuroimaging manifestations in COVID-19-associated encephalitis. From January 2020 to September 2021, patients with clinical diagnosis of COVID-19-associated encephalitis, as well as concomitant abnormal imaging findings on brain MRI, were included. Two board-certified neuro-radiologists reviewed these selected brain MR images, and further discerned the abnormal imaging findings. 39 patients with the clinical diagnosis of encephalitis as well as abnormal MRI findings were included. Most (87%) of these patients were managed in ICU, and 79% had to be intubated-ventilated. 15 (38%) patients died from the disease, while the rest were discharged from the hospital. On MRI, FLAIR hyperintensities in the insular cortex were the most common finding, seen in 38% of the patients. Micro-hemorrhages on the SWI images were equally common, also seen in 38% patients. FLAIR hyperintensities in the medial temporal lobes were seen in 30%, while FLAIR hyperintensities in the posterior fossa were evident in 20%. FLAIR hyperintensities in basal ganglia and thalami were seen in 15%. Confluent FLAIR hyperintensities in deep and periventricular white matter, not explained by microvascular angiopathy, were detected in 7% of cases. Cortical-based FLAIR hyperintensities in 7%, and FLAIR hyperintensity in the splenium of the corpus callosum in 7% of patients. Finally, isolated FLAIR hyperintensity around the third ventricle was noted in 2% of patients.

**Keywords:** COVID-19; magnetic resonance imaging; encephalitis

## 1. Introduction

SARS-CoV-2 outbreak was first reported in Wuhan and has continued to cause devastation across the globe, with more than 6.3 million reported deaths to date [1]. New strains have emerged which have infected even vaccinated persons. Respiratory, hematologic, cardiac, and gastrointestinal complications of COVID-19 have been extensively studied. Although COVID-19 primarily affects the respiratory system, studies have found that it also adversely impacts the neurological system. The most commonly reported neurological symptoms include fatigue, myalgia, taste alteration, smell impairment, and headache [2,3]. Central nervous system (CNS) manifestations include stroke, cerebrovascular thromboembolism, anosmia, and myelitis. Thrombogenic complications of the disease, such as ischemic stroke and dural venous thrombosis, are well-known [4]. Limited studies and case reports have also reported non-hemorrhagic and hemorrhagic leukoencephalitis as potential complications of COVID-19 [5].

Encephalitis is defined as CNS inflammation mediated by viral and, less commonly, bacterial infectious agents or immune reactions, and presents with symptoms of altered mental status (AMS), seizures, and fever [6]. Encephalitis is a rare manifestation of COVID infection with an incidence of 0.2–1%, but a poor prognosis. It has been reported that the incidence of COVID encephalitis in severely ill patients is about 6–7% [7]. Encephalitis usually presents 14 days after the onset of COVID symptoms. About 83% of patients

with encephalitis are already admitted to the intensive care unit (ICU). The mean age of COVID-associated encephalitis is 59 years without a gender preference. Hypertension, hyperlipidemia, and diabetes mellitus are the most common comorbidities. The most frequent manifestation of COVID-associated encephalitis is decreased level of consciousness, altered mental status, seizures, headaches, and weakness [7]. The mechanism of COVID-associated encephalitis is not known. Possible mechanisms of CNS inflammation in these patients include autoimmune reaction to the anti-NMDA (N-methyl D-aspartate) receptor antibodies against the glutamatergic pathways [8], inflammation of the glial system [9,10], or ischemic changes caused by intravascular thrombosis given the presence of ACE2 receptors in the vascular endothelium [11]. Therefore, diagnosing COVID-19-induced encephalitis remains a challenge due to a variable range of symptoms, neuroimaging findings, and lack of an established diagnostic criteria. Detection of SARS-CoV2 in the CSF remains insufficient [12].

Therefore, this retrospective study explored the spectrum of the neuroimaging manifestations of COVID-associated encephalitis.

## 2. Materials and Methods

This study was approved by the local ethical committee of our tertiary care institution (IRB300008377-0022, 8 November 2021) in United States. After scrutinizing the radiology reports for magnetic resonance imaging (MRI) studies performed on COVID-19 patients from January 2020 to September 2021, all patients with clinical diagnosis of COVID-associated encephalitis were included. Exclusion criteria included non-COVID-19-encephalitis pathologies, such as acute or chronic infarcts, intracranial malignancy, and known encephalitis due to other causes such as infection, metabolic derangements, and inflammatory causes. The diagnosis of COVID-19 encephalitis was established using modified diagnostic criteria combining the clinical diagnosis by the primary care team, diagnostic criteria for encephalitis [13] (Table 1), and a mandatory presence of positive COVID-19 polymerase chain reaction (PCR) test. Patients with an encephalitis diagnosis from the primary care team, however, not meeting the criteria based on the diagnostic criteria for encephalitis were excluded. All MRI imaging was performed on 3 T and 1.5 T Siemens and Philips MRI scanners at our university hospital affiliated with a medical school. Patients' clinical information, such as age, gender, co-morbidities, presence of AMS, fever, seizure, EEG findings, initial presenting symptoms, presence of neurological deficits or psychiatric symptoms, reason for MRI, and days from admission to MRI were recorded from the Cerner powerchart (Kansas City, Missouri). Two board-certified neuro-radiologists then reviewed the brain MRIs of the selected patients regarding the presence of abnormal imaging findings. Both neuroradiologists being at separate workstations simultaneously reviewed these cases over Zoom video conference (San Jose, California), using Philips IntelliSpace PACS (Amsterdam, The Netherlands) to ensure appropriate social distancing during the pandemic. Image findings were recorded after consensus. T1-weighted images were utilized for screening structural abnormalities. We reviewed the FLAIR (fluid-attenuated inversion recovery), SWI (susceptibility-weighted images), DWI (diffusion-weighted images), ADC (apparent diffusion coefficient), and post-contrast (if available) MRI sequences for any abnormal findings of hemorrhagic or non-hemorrhagic encephalitis. Since the majority of patients did not undergo contrast enhanced imaging, we excluded the post-contrast imaging findings. Studied findings include microhemorrhages, macroscopic hemorrhages, and FLAIR hyperintensities. FLAIR signal characteristics of the bilateral frontal lobe cortices were used as a comparative standard for assessing the FLAIR hyperintensity. DWI images were primarily utilized for the exclusion of acute and subacute infarcts. In addition, we also recorded the location of the abnormality: medial temporal lobe, other specific parenchymal lobes, diffuse presence, insula, basal ganglia, thalamus, corpus callosum, periventricular location, middle cerebellar peduncle, cerebellum, and brainstem. Furthermore, we also recorded ICU admission, need of ventilation, treatment received, days in the hospital, and final outcome for all participants.

Table 1. Modified Diagnostic Criteria for COVID-19 Encephalitis [13].

| Modified Diagnostic Criteria for COVID-19 Encephalitis |
| --- |
| **Major Criterion (required)** |
| Patient must have a positive COVID-19 PCR test |
| Patients presenting to medical attention with altered mental status (defined as decreased or altered level of consciousness, lethargy or personality change) lasting >= 24 h with no alternative cause identified |
| **Minor Criteria (2 required for possible encephalitis; >= 3 required for probable or confirmed encephalitis)** |
| Documented fever >= 38 C (100.4 F) within the 72 h before or after presentation |
| Generalized or partial seizures not fully attributable to a preexisting seizure disorder |
| New onset of focal neurologic findings |
| CSF WBC count >= 5/cubic mm |
| Abnormality of brain parenchyma on neuroimaging suggestive of encephalitis that is either new from prior studies or appears acute in onset |
| Abnormality on electroencephalography that is consistent with encephalitis and not attributable to another cause |
| Abbreviations: CSF-cerebral spinal fluid, PCR-polymerase chain reaction, WBC-white blood cell |

## 3. Results

During the study time frame, a total of 527 COVID-19 patients underwent brain MRI, from which patients with stroke, malignancy, PRES, and other non-COVID-19 encephalitis pathologies were excluded. 71 patients met the clinical criteria of COVID-19 encephalitis, out of which 32 patients had a normal brain MRI. Eventually, 39 patients with abnormal brain MRI findings (21 female and 18 male; mean age 61 years) were selected for the final analysis (Figure 1).

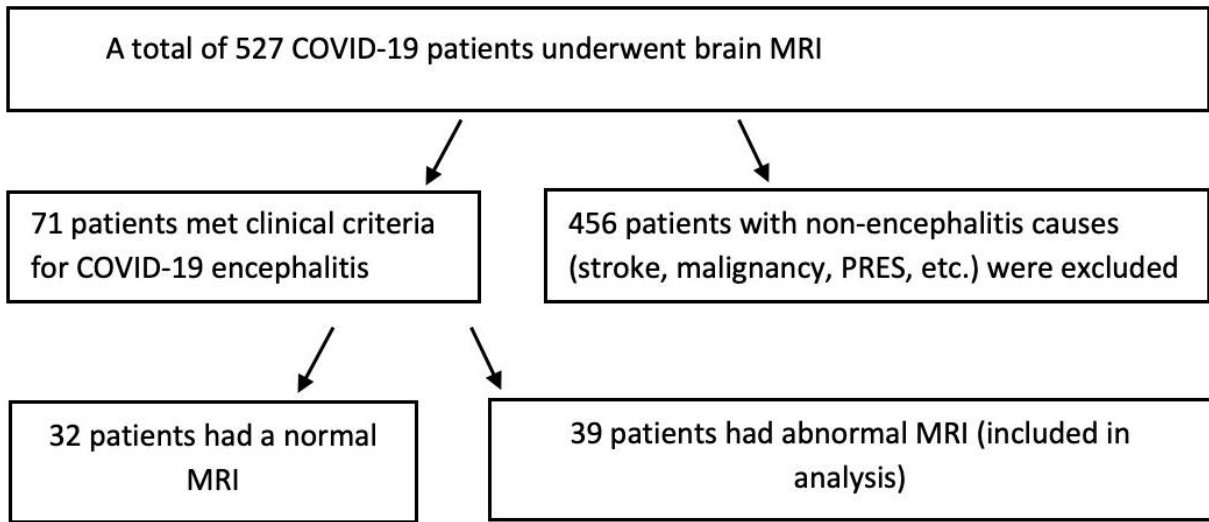

**Figure 1.** Patient enrollment flow chart.

The clinical characteristics of these patients are described in Table 2. The most common initial presenting symptoms were fever, cough, shortness of breath, altered mental status, and loss of consciousness. The majority of patients did not report psychiatric symptoms, however, 10/39 patients (26%) reported neurologic deficits. 34/39 (87%) patients were admitted to the intensive care unit (ICU), and 31/39 (79%) had to be intubated-ventilated. MRI scan was performed between 1–90 days after admission (mean = 16 days), while hospital stay ranged from 4–150 days (mean = 36 days). 18 (46%) patients received remdesivir, while 2 (5%) were given acyclovir. A total of 15 (38%) patients died from the disease, while the rest were discharged from the hospital.

The imaging characteristics are described in Table 3. FLAIR hyperintensities in the insular cortex were the most common neuroimaging manifestation of the COVID-induced encephalitis, seen in 15 patients (38%), with a bilateral pattern evident in ten of them (25%) (Figure 2A). Isolated left insular involvement was seen in 5 patients (12)% (Figure 2B). We did not detect isolated right insular involvement. Cerebral microhemorrhages on the SWI sequence were equally prevalent manifestation in 15 patients (38%), with a diffuse pattern in 9 patients (23%) (Figure 3A), cerebellar micro-hemorrhages in 2 patients (5%) (Figure 3B), frontal micro-hemorrhages in 2 patients (5%) (Figure 3C), 1 each in left basal ganglia (2%) and corpus callosum splenium (2%).

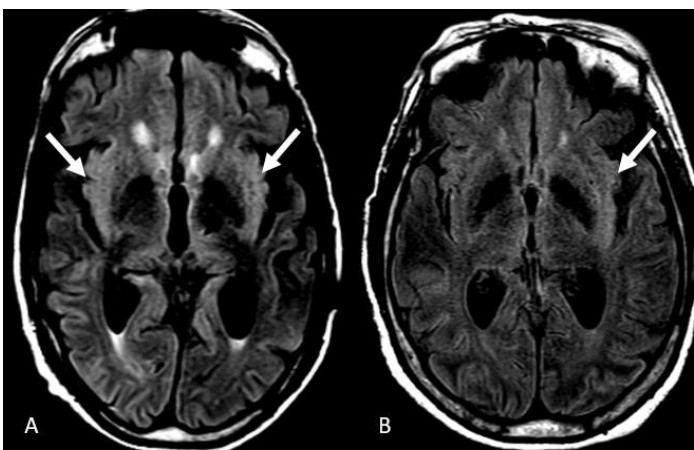

**Figure 2.** Insular involvement. (**A**) Axial FLAIR images demonstrate bilateral FLAIR hyperintensities in the insular cortex. (**B**) isolated left insular involvement. Image has been windowed manually to improve the visibility of signal abnormality.

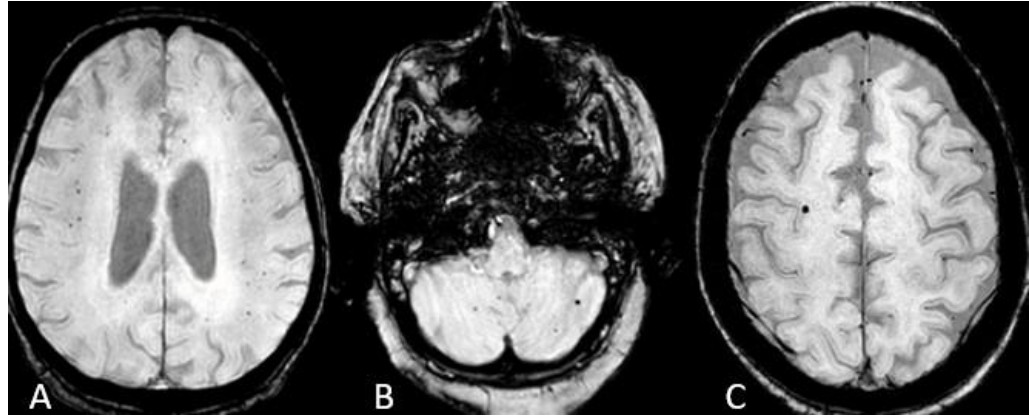

**Figure 3.** Microhemorrhages. (**A**) SWI images show the diffuse pattern of microhemorrhages. (**B**) Isolated cerebellar microhemorrhages. (**C**) Isolated frontal microhemorrhage.

FLAIR hyperintensities in the medial temporal lobes were the third most common presentation reported in 12 patients (30%). These findings were bilateral in 10 patients (25%) (Figure 4A) and isolated left-sided in 2 patients (5%) (Figure 4B). We did not detect isolated right temporal FLAIR hyperintensities. FLAIR hyperintensities in the posterior fossa were the fourth most common manifestation of COVID-19-associated encephalitis evident in 9 patients (20%), which in 6 of them effected the ventral pons (15%) (Figure 5A), bilateral middle cerebellar peduncles in 1 patient (Figure 5B), vermis in 1 patient (Figure 5C), and bilateral cerebellar hemispheres in 1 patient (Figure 5D). FLAIR hyperintensities in the basal ganglia and thalami were the fifth most common finding evident in 6 patients (15%), with a bilateral involvement pattern in 4 patients (10%) (Figure 6A), and isolated bilateral thalami findings in 2 patients (5%) (Figure 6B).

**Table 2.** Clinical Characteristics of the Participants.

| ID | Age | Sex | Comorbidities | AMS > 24 h | T >= 38, 100.4 F | Generalized or Partial Seizure | EEG | Initial Presenting Symptoms | Neural Deficits | Psychiatric Symptoms | Admission to MRI (in Days) | Reason for MRI | ICU Admission | Ventilated? | Treatment | Days in Hospital | Outcome |
|----|-----|-----|---------------|-----------|-------------------|-------------------------------|-----|-----------------------------|-----------------|----------------------|----------------------------|----------------|---------------|-------------|-----------|-------------------|---------|
| 1 | 51 | M | Hemorrhagic stroke, HTN, DM, BPH, HLD, gout and impaired mobility | yes | yes | yes | yes | AMS, encephalitis | no | no | 3 | Encephalitis and seizure | yes | yes | Dexamethasone, remdesivir | 27 | Discharge |
| 2 | 42 | F | ESRD on PD, CVA, DM2, and blindness | yes | yes | no | no | Fever, cough, weakness | yes | no | 1 | Confusion | yes | yes | Dexamethasone | 28 | Death |
| 3 | 49 | M | Type II DM | yes | yes | yes | yes | Fever, cough, AMS | no | no | 19 | Encephalitis | yes | yes | Dexamethasone | 38 | Discharge |
| 4 | 82 | F | Type I diabetes mellitus, hypertension | yes | yes | yes | yes | Confusion, acute onset aphasia, encephalitis | yes | no | 1 | Aphasia, encephalitis, facial droop, visual field loss | yes | yes | Dexamethasone, remdesivir | 17 | Discharge |
| 5 | 62 | F | Renal transplant, recurrent UTIs, CKD, HTN, DM | yes | yes | no | no | AMS | no | no | 9 | AMS, encephalitis | yes | yes | Dexamethasone | 17 | Death |
| 6 | 68 | F | NASH cirrhosis, hepatic encephalopathy, DM, HLD, pancreatitis, OSA, HTN | yes | yes | no | no | Shortness of breath, cough, confusion | no | no | 13 | Encephalitis, AMS | yes | yes | Dexamethasone, remdesivir | 28 | Discharge |
| 7 | 62 | M | HTN, type II DM, OSA, rectal fissure, inhalation injury | yes | yes | yes | no | Fatigue, weakness, cough, dyspnea, | no | no | 61 | Encephalitis | yes | yes | Dexamethasone, remdesivir | 65 | Discharge |
| 8 | 68 | M | Uncontrolled HTN, vascular dementia | yes | yes | no | no | Hypertensive emergency | no | no | 7 | Encephalitis and Parkinsonian symptoms | yes | yes |  | 57 | Discharge |
| 9 | 59 | M | Morbid obesity, NIDDM, HTN, gout, GERD/PUD, OSA, retinal vein occlusion | yes | yes | yes | yes | Fever, cough, progressive dyspnea | yes | no | 36 | Encephalitis, hydrocephalus | yes | yes | Dexamethasone, remdesivir | 49 | Discharge |

**Table 2.** *Cont.*

| ID | Age | Sex | Comorbidities | AMS > 24 h | T >= 38, 100.4 F | Generalized or Partial Seizure | EEG | Initial Presenting Symptoms | Neural Deficits | Psychiatric Symptoms | Admission to MRI (in Days) | Reason for MRI | ICU Admission | Ventilated? | Treatment | Days in Hospital | Outcome |
|---|---|---|---|---|---|---|---|---|---|---|---|---|---|---|---|---|---|
| 10 | 59 | F | DVT (on Xarelto), HTN, sickle cell disease, CKD | yes | yes | no | yes | Syncopal episode with residual weakness, L foot and neck pain | yes | no | 6 | Encephalitis, follow up stroke | yes | yes | Dexamethasone | 15 | Discharge |
| 11 | 53 | F | DM type I, renal transplant, neuropathy, hypothyroidism, HTN | yes | yes | no | no | Generalized body aches, decreased UOP and fever | no | no | 10 | Encephalitis | yes | yes | Dexamethasone, remdesivir | 46 | Discharge |
| 12 | 84 | F | A-fib and HTN | yes | yes | yes | yes | Shortness of breath, cough | no | no | 9 | AMS, possible infarcts seen on CT | yes | yes | Dexamethasone, remdesivir | 39 | Discharge |
| 13 | 66 | F | HTN, undiagnosed diabetes | yes | yes | yes | yes | AMS, unresponsiveness | yes | no | 4 | Encephalitis, AMS | yes | no | Dexamethasone, remdesivir | 28 | Discharge |
| 14 | 65 | F | HTN, diabetes, hyperlipidemia | yes | yes | yes | yes | Shortness of breath, encephalitis | yes | no | 23 | Encephalitis, AMS, PRES | yes | no | Dexamethasone, remdesivir | 36 | Discharge |
| 15 | 42 | F | IDDM, HTN | yes | yes | yes | yes | Generalized tonic–clonic seizures | no | no | 2 | Encephalitis, AMS, seizure | yes | yes | Dexamethasone | 6 | Death |
| 16 | 77 | F | HTN, gout | yes | yes | no | no | AMS, falls | no | no | 5 | Encephalitis, confusion | yes | no | Dexamethasone, remdesivir | 39 | Death |
| 17 | 78 | M | CAD s/p CABG, Parkinson's L STN stimulator lead, HLD, HTN | yes | yes | yes | yes | Cough, progressive dyspnea, pleuritic chest pain | yes | no | 25 | Encephalitis | yes | yes | Dexamethasone, remdesivir | 34 | Death |
| 18 | 78 | F | psychotic disorder, HTN, diabetes | yes | yes | no | no | Hypoxia and lethargy | yes | no | 10 | AMS | yes | yes | Dexamethasone | 24 | Discharge |
| 19 | 32 | F | Insulin-dependent diabetes, heart failure, ESRD | yes | yes | yes | yes | Shortness of breath, cough and fever. | no | no | 6 | Status post PEA arrest | yes | yes | Dexamethasone | 14 | Death |

**Table 2.** *Cont.*

| ID | Age | Sex | Comorbidities | AMS > 24 h | T >= 38, 100.4 F | Generalized or Partial Seizure | EEG | Initial Presenting Symptoms | Neural Deficits | Psychiatric Symptoms | Admission to MRI (in Days) | Reason for MRI | ICU Admission | Ventilated? | Treatment | Days in Hospital | Outcome |
|---|---|---|---|---|---|---|---|---|---|---|---|---|---|---|---|---|---|
| 20 | 58 | F | Hypertension, diabetes, schizophrenia, hepatitis C | yes | yes | no | yes | Dyspnea, progressive lethargy, fever, loss of appetite and hypotension | no | no | 21 | Encephalitis, AMS | yes | yes | Dexamethasone, remdesivir | 30 | Discharge |
| 21 | 82 | F | HTN, DM | yes | yes | yes | yes | Loss of consciousness | no | no | 9 | AMS | yes | yes | Dexamethasone, remdesivir | 16 | Death |
| 22 | 77 | M | CAD, aortic valve stenosis | yes | yes | no | yes | Hypoxia after surgery for TAVR | no | no | 36 | Encephalitis, AMS | yes | yes | | 59 | Death |
| 23 | 60 | F | HLD, DM, stroke, AML, migraine, depression, spinal stenosis | yes | yes | no | yes | Loss of consciousness | no | no | 10 | AMS, weakness | yes | yes | | 37 | Discharge |
| 24 | 41 | M | MS | yes | yes | no | yes | Fever, blurred vision | yes | no | 1 | Ataxia, blurred vision | no | no | | 5 | Discharge |
| 25 | 66 | F | Sarcoidosis, lung transplant | yes | yes | no | yes | Shortness of breath | no | no | 30 | Encephalitis | no | no | Dexamethasone, remdesivir | 40 | Death |
| 26 | 55 | F | HTN, DM, CSF leak, gout, fibromyalgia | yes | yes | yes | yes | Fever, cough, AMS | no | no | 7 | Encephalitis | no | no | Dexamethasone | 20 | Discharge |
| 27 | 45 | M | HTN, CKD, obesity | yes | yes | no | yes | Hypoxia | no | no | 17 | Encephalitis | yes | yes | Dexamethasone, remdesivir | 56 | Discharge |
| 28 | 46 | M | Obesity, gout, | yes | yes | no | yes | Hypoxia | no | no | 21 | Encephalitis | yes | yes | Acyclovir | 43 | Death |
| 29 | 69 | F | COPD, sarcoidosis, schizoaffective, DM, HTN | yes | yes | no | yes | Fever | no | no new | 13 | Encephalitis | yes | yes | Dexamethasone, remdesivir | 26 | Death |
| 30 | 41 | M | None | yes | yes | no | yes | Weakness | no | no | 23 | AMS | yes | yes | Dexamethasone, remdesivir | 31 | Death |
| 31 | 41 | F | CHF, DM, GERD, | yes | yes | no | yes | Fever | no | no | 90 | Encephalitis | yes | yes | | 150 | Death |
| 32 | 56 | M | CVA, TIA, depression, anemia, HTN, DM | yes | yes | yes | yes | Seizure | no | no | 2 | Encephalitis, seizure | yes | yes | Acyclovir | 5 | Discharge |
| 33 | 72 | M | DM, DAC, | yes | yes | no | yes | Hypoxia | no | no | 34 | Encephalitis | yes | yes | | 45 | Death |

**Table 2.** *Cont.*

| ID | Age | Sex | Comorbidities | AMS > 24 h | T >= 38, 100.4 F | Generalized or Partial Seizure | EEG | Initial Presenting Symptoms | Neural Deficits | Psychiatric Symptoms | Admission to MRI (in Days) | Reason for MRI | ICU Admission | Ventilated? | Treatment | Days in Hospital | Outcome |
|---|---|---|---|---|---|---|---|---|---|---|---|---|---|---|---|---|---|
| 34 | 77 | M | HTN, DM, CHF, HTN, prostate CA | yes | no | no | yes | Hypoxia | no | no | 12 | Encephalitis | yes | yes | | 47 | Discharge |
| 35 | 61 | M | HTN, TBI | yes | no | yes | yes | Seizure | yes | no | 13 | Seizure | no | no | Dexamethasone, remdesivir | 90 | Discharge |
| 36 | 63 | M | DM, COPD, ESRD | yes | yes | yes | yes | Hypoxia | no | no | 18 | AMS | yes | yes | Dexamethasone | 24 | Death |
| 37 | 83 | M | HTN, HLD, DM | yes | no | yes | yes | Loss of consciousness | no | no | 4 | AMS | no | no | | 4 | Discharge |
| 38 | 47 | F | DM, HTN, obesity | yes | yes | no | yes | Hypoxia | no | no | 9 | AMS | yes | yes | | 45 | Discharge |
| 39 | 68 | M | DM, HTN, obesity | yes | no | no | yes | Shortness of breath | no | no | 23 | Encephalitis | yes | yes | | 33 | Discharge |

**Table 3.** Imaging Characteristics of the Participants.

| ID | Age | Sex | FLAIR Hyperintensities | | | | | | | | | Microhemorrhages |
|---|---|---|---|---|---|---|---|---|---|---|---|---|
| | | | Insula | Medial Temporal Lobe | Ventral Pons | Cerebellum | Deep Gray Matter | Cortical Signal Abnormality | Confluent Leukoencephalopathy | Around Third Ventricle | Corpus Callosum Splenium | |
| 1 | 51 | M | | | Yes | | | | | | | |
| 2 | 42 | F | | | | | | Right occipital | | | | |
| 3 | 49 | M | B/l | B/l | Yes | | | B/l occipital and pareital | | | | Cerebellar |
| 4 | 82 | F | Left | | | | | | | | | |
| 5 | 62 | F | Left | | | | | | | | | Right frontal, right cerebellum |
| 6 | 68 | F | B/l | B/l | Yes | | B/l basal ganglia & thalami | | | | | |
| 7 | 62 | M | | | | | B/l basal ganglia & thalami | | | | | |
| 8 | 68 | M | | | | | | | | | | Multiple |
| 9 | 59 | M | | | Yes | | | | | | | |
| 10 | 59 | F | | | | | | | | Yes | | |
| 11 | 53 | F | B/l | | Yes | | | | | | | |
| 12 | 84 | F | B/l | B/l | | | | | | | | |
| 13 | 66 | F | | | | | | | | | | Multiple |

**Table 3.** *Cont.*

| ID | Age | Sex | FLAIR Hyperintensities | | | | | | | | | Microhemorrhages |
|---|---|---|---|---|---|---|---|---|---|---|---|---|
| | | | Insula | Medial Temporal Lobe | Ventral Pons | Cerebellum | Deep Gray Matter | Cortical Signal Abnormality | Confluent Leukoen-cephalopathy | Around Third Ventricle | Corpus Callosum Splenium | |
| 14 | 65 | F | B/l | B/l | | | | | | | | |
| 15 | 42 | F | | | | | B/l basal ganglia & thalami | | | | | |
| 16 | 77 | F | | | | | | | | | | Multiple |
| 17 | 78 | M | B/l | B/l | | | B/l thalami | | | | | |
| 18 | 78 | F | | | | | | | | Yes | Yes | |
| 19 | 32 | F | | | | | | | | | | Left basal ganglia |
| 20 | 58 | F | | | | | | | Yes | | | |
| 21 | 82 | F | | | | | | | | | Yes | |
| 22 | 77 | M | | | | | B/l basal ganglia & thalami | | | | | Frontal |
| 23 | 60 | F | B/l | B/l | | | | | | | | |
| 24 | 41 | M | | B/l | Yes | | | | Yes | | | |
| 25 | 66 | F | | | | | | | | | | Multiple |
| 26 | 55 | F | | | | | | B/l parietal | | | | |
| 27 | 45 | M | B/l | B/l | | | | | | | | |
| 28 | 46 | M | | | | Vermis | | | | | | |
| 29 | 69 | F | Left | | | | | | | | | Multiple |
| 30 | 41 | M | | Left | | | | | | | | Corpus callosum splenium |
| 31 | 41 | F | B/l | B/l | | | | | | | | |
| 32 | 56 | M | | | | B/l | | | | | | Multiple |
| 33 | 72 | M | | | | | | | | | | Multiple |
| 34 | 77 | M | | | B/l middle cerebellar peduncles | | | | | | Yes | |
| 35 | 61 | M | B/l | B/l | | | | | | | | |
| 36 | 63 | M | | | | | B/l thalami | | | | | Multiple |
| 37 | 83 | M | Left | | | | | | | | | |
| 38 | 47 | F | | | | | | | | | | Multiple |
| 39 | 68 | M | Left | Left | | | | | | | | |

Abbreviations: B/l—bilateral.

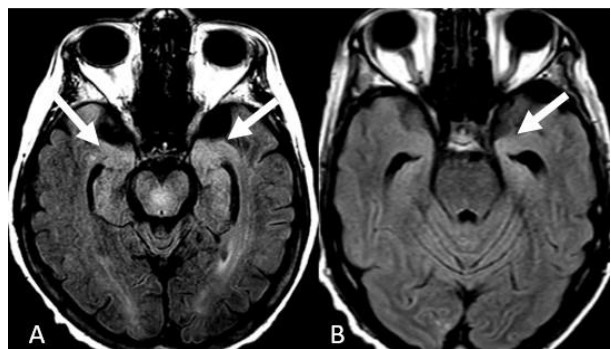

**Figure 4.** Temporal lobe involvement. (**A**) Axial FLAIR images depict bilateral medial temporal lobe FLAIR hyperintensities. (**B**) Isolated left medial temporal FLAIR hyperintensities. Image has been windowed manually to improve the visibility of signal abnormality.

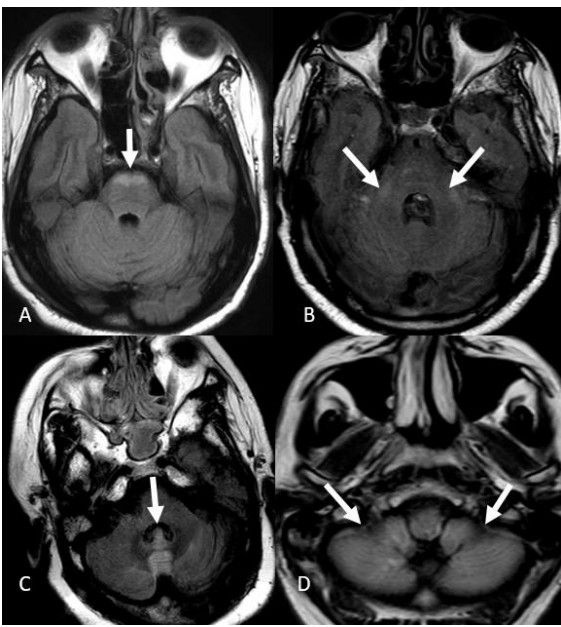

**Figure 5.** Posterior fossa involvement. (**A**) Axial FLAIR images demonstrate FLAIR hyperintensity in the ventral portion of the pons. (**B**) Bilateral middle cerebellar peduncle FLAIR hyperintensities. (**C**) FLAIR hyperintensity of the vermis. (**D**) FLAIR hyperintensities in the bilateral cerebellar hemispheres. Image has been windowed manually to improve the visibility of signal abnormality.

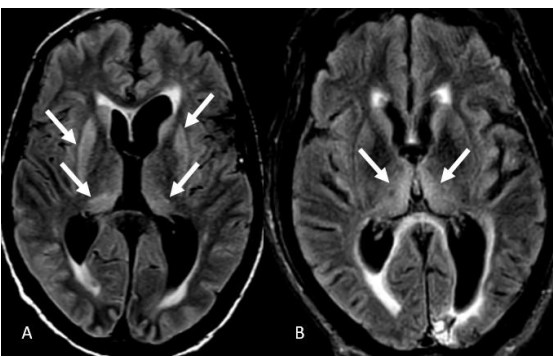

**Figure 6.** Basal ganglia and thalami involvement. (**A**) Axial FLAIR images show bilateral basal ganglia and thalami FLAIR hyperintensities. (**B**) Bilateral thalami FLAIR hyperintensities. Image has been windowed manually to improve the visibility of signal abnormality.

In some cases, confluent patchy FLAIR hyperintensities in the deep and periventricular white matter (diffuse leukoencephalopathy) could not be explained by chronic microvascular angiopathic changes, and were considered to be the sixth manifestation in 3 patients (7%) (Figure 7). Cortical-based FLAIR hyperintensities were seen in 3 patients (7%), involving the right occipital lobe in one of the patients (2%) (Figure 8A), bilateral parieto-occipital lobes in 1 patient (2%) (Figure 8B), and bilateral occipital lobes in 1 patient (2%) (Figure 8C). Less common patterns include FLAIR hyperintensities in the splenium of the corpus callosum observed in 3 patients (7%). (Figure 9A). Finally, isolated FLAIR hyperintensities around the third ventricle were seen in 1 patient (2%) (Figure 9B).

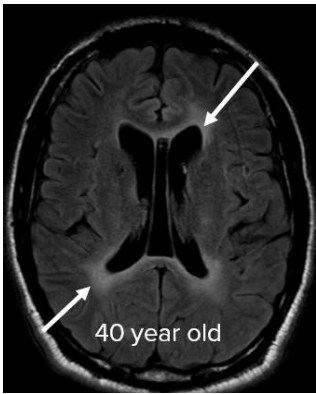

**Figure 7.** Periventricular involvement. Axial FLAIR image depicts periventricular FLAIR hyperintensities in a 40-year-old patient, otherwise not explained by chronic microangiopathic changes and absence of other risk factors. Image has been windowed manually to improve the visibility of signal abnormality.

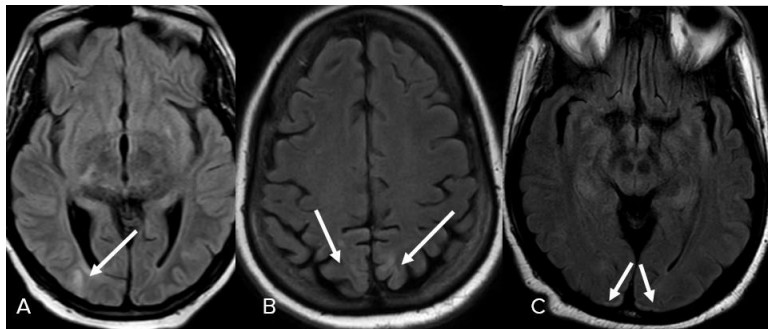

**Figure 8.** Cortical involvement. (**A**) Axial FLAIR images demonstrate isolated cortical-based FLAIR hyperintensity in the right occipital lobe. (**B**) Bilateral parietal FLAIR hyperintensities. (**C**) Bilateral occipital FLAIR hyperintensities. Image has been windowed manually to improve the visibility of signal abnormality.

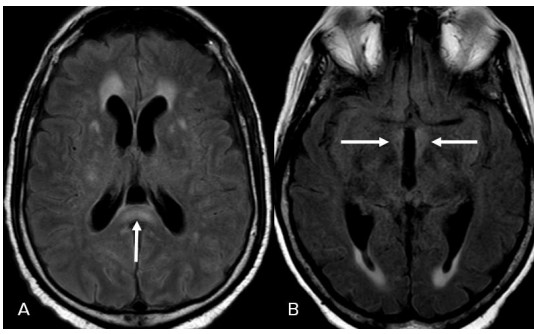

**Figure 9.** Less common patterns. (**A**) Axial FLAIR images show FLAIR hyperintensity of the splenium of the corpus callosum. (**B**) FLAIR hyperintensity around the third ventricle. Image has been windowed manually to improve the visibility of signal abnormality.

## 4. Discussion

We conducted this study to navigate the unknown scope of neuroimaging manifestations in COVID-associated encephalitis. After extensive research on the patient's medical records, 39 patients with a mean age of 61 years were enrolled in the study. The results demonstrated that abnormal FLAIR hyperintensities in the insular cortex were the most common radiologic findings in COVID-associated encephalitis, reported in 38% of the patients. In addition, micro-hemorrhages evident on the SWI sequences were equally prevalent. Moreover, FLAIR hyperintensities in the medial temporal lobes were observed in 30%, which was the third most common presentation. It is also worth noting that FLAIR hyperintensities in the posterior fossa, basal ganglia and thalami accounted for 20% and 15% of cases, respectively. Finally, this data suggests that diffuse leukoencephalopathy, abnormal signal intensity in the splenium of the corpus callosum, and periventricular white matter signal changes around the third ventricle may be the other less common manifestations.

Our results are consistent with Kandemirli et al. study, which suggested that one of the most common imaging finding was cortical signal abnormalities on FLAIR images [14]. Presence of FLAIR abnormalities and microhemorrhages has been reported in other studies as well [5,15–17]. Some studies have proposed leptomeningeal enhancement; however, majority of our patients did not undergo contrast-enhanced images. Furthermore, some articles have described the presence of restriction diffusion in the white matter; however, we primarily utilized DWI images to exclude cases with acute and subacute infarcts to avoid overlap with ischemic changes secondary to thrombotic complications [18,19].

MRI findings in critically ill or ICU-admitted COVID-19 patients have been previously published by many researchers. The precise pathophysiology is unclear; however, authors have hinted towards immunologic parainfectious processes [20], which has been supported by some neuropathologic studies [21]. Radmanesh et al. evoked the assumptions of hypoxia or small-vessel vasculitis [16]. Direct brain tissue involvement by viral particles has not been endorsed by cerebrospinal fluid analysis, and other hypotheses such as postinfectious demyelinating diseases may be considered. Since most of these patients were admitted to intensive care units for acute respiratory distress syndrome, other assumptions include delayed post hypoxic leukoencephalopathy [22], metabolic or toxic encephalopathy, and posterior reversible encephalopathy syndrome [23].

While differentiation from other causes of encephalitis can be difficult, other etiologies such as herpes encephalitis present with a fulminant course, with 90% of cases resulting from HSV-1. Diagnosis is established with CSF PCR, although a combination of the clinical scenario, CSF pleocytosis and elevated protein, and appropriate imaging are usually highly suggestive and permit commencement of treatment [24]. Similarly, autoimmune encephalitis can present with the involvement of the mesial temporal lobes and limbic system, manifested by cortical thickening and increased T2/FLAIR signal intensity. However, in approximately 60% of cases, antineuronal antibodies are present, such as the anti-Hu antibody in small cell lung cancer, anti-Ta antibody in testicular cancers, anti-NMDA NR1 in ovarian teratomas, or anti-NMDA NR2 in SLE patients [25].

The current study is one of the most extensive research studies by sample size, assessing imaging findings of clinically documented encephalitis in COVID-19 patients. However, we may have encountered some limitations. The study's retrospective nature may affect its reliability in some cases and limit prospective application. Additionally, although this study was conducted at a large tertiary care center, pooling data from multiple institutions will add to the generalizability of the results. Additionally, post-contrast sequences were not evaluated in our study but may reveal additional useful imaging features in COVID-associated encephalitis. As a result, further multi-institutional investigations and also utilization contrast-enhanced imaging can be considered for future studies.

## 5. Conclusions

COVID-associated encephalitis manifests as a spectrum of neuroimaging findings on brain MRI. The most common manifestations are abnormal FLAIR hyperintensities in the

insular cortex, and diffuse or focal microhemorrhages. Other neuroimaging manifestations are abnormal FLAIR hyperintensities in the medial temporal lobes, brainstem, abnormal cortical hyperintensities, and abnormal hyperintensities within the thalami and basal ganglia. Other less common findings include diffuse leukoencephalopathy, abnormal FLAIR hyperintensities in the corpus callosum splenium, and around the third ventricle.

**Author Contributions:** Conceptualization, M.T. and H.S.; methodology, M.T. and H.S.; validation, M.T., A.S. and H.S.; formal analysis, M.T., A.S., M.A. and H.S.; investigation, M.T., A.S, M.A. and H.S.; data curation, M.T., M.A and H.S; writing— original draft, M.T. and H.S; writing—review and editing, M.T., A.S., M.A. and H.S; visualization, M.T., A.S., M.A. and H.S.; supervision, M.T.; project administration, M.T and H.S. All authors have read and agreed to the published version of the manuscript.

**Funding:** This research received no external funding.

**Institutional Review Board Statement:** This study was approved by the institutional review board of the University of Alabama at Birmingham (IRB300008377-0022, 8 November 2021).

**Informed Consent Statement:** Patient consent was waived due to the retrospective chart review nature of the study. No additional risk to the subjects.

**Data Availability Statement:** The clinical, imaging findings data and anonymized images are already published in the manuscript. Source MRI images belong to specific patients, which are stored in the local radiology PACS, and are not available for public sharing. Any further inquiry can be directed to the authors.

**Acknowledgments:** We thank our MRI technologists, including our inpatient MRI head technologist, Jeff Madison, for their continued service in scanning these ill patients throughout the COVID-19 pandemic.

**Conflicts of Interest:** The authors declare no conflict of interest.

## Abbreviations

ADC = apparent diffusion coefficient, ACE = angiotensin-converting enzyme, AMS = altered mental status, CNS = central nervous system, COVID-19 = coronavirus disease 2019, DWI = diffusion-weighted images, FLAIR = fluid-attenuated inversion recovery, ICU = intensive care unit, GCS = Glasgow coma scale, MRI = magnetic resonance imaging, NIHSS = NIH stroke scale, NMDA = N-methyl D-aspartate, PACS = picture archiving and communication system, PCR = polymerase chain reaction, SARS-CoV-2 = severe acute respiratory syndrome coronavirus 2, SWI = susceptibility-weighted images.

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
