# Peer review of "Magnetic Resonance Imaging (MRI) Findings in COVID-19 Associated Encephalitis"

_2035-8377, doi:10.3390/neurolint15010005_

Round 1
Reviewer 1 Report
Authors work is one of the few that analyzes brain MRI findings in COVID19 patients. Particularly, they focused their research on encephalitis, a serious neurologic complication of COVID19. I have some major and minor considerations about their work.
As major concerns:
- Why authors included just patients with encephalitis and altered brain MRI findings? From the criteria showed in table 1, MRI suggestive findings are not necessary for the diagnosis. I believe that even data on the percentage of normal MRI findings are relevant.
- Discussion is a mere description of results, I believe that authors should add their considerations and try to find hypotheses to explain the MRI findings.
As minor concern:
- a table showing patients demographic, clinical characteristics, and comorbidities should be added.
Author Response
Authors work is one of the few that analyzes brain MRI findings in COVID19 patients. Particularly, they focused their research on encephalitis, a serious neurologic complication of COVID19. I have some major and minor considerations about their work.
As major concerns:
- Why authors included just patients with encephalitis and altered brain MRI findings? From the criteria showed in table 1, MRI suggestive findings are not necessary for the diagnosis. I believe that even data on the percentage of normal MRI findings are relevant.
>> Thank you for the comment. In our study, 79 patients with COVID-19 encephalitis underwent MRI, out of these 32 had normal MRIs. 39 patients had abnormal MRI findings. This data has been added.
- Discussion is a mere description of results, I believe that authors should add their considerations and try to find hypotheses to explain the MRI findings.
>> Thank you for the comment. Added paragraph regarding hypotheses for these MRI abnormalities.
MRI findings in critically ill or ICU-admitted COVID-19 patients have been previously published by many researchers. The precise pathophysiology is unclear; however, authors have hinted towards immunologic parainfectious processes20, which has been supported by some neuropathologic studies21. Radmanesh et al evoked the assumptions of hypoxia or small-vessel vasculitis16. Direct brain tissue involvement by viral particles has not been endorsed by cerebrospinal fluid analysis, and other hypotheses such as postinfectious demyelinating diseases may be considered. Since most of these patients were admitted to intensive care units for acute respiratory distress syndrome, other assumptions include delayed post hypoxic leukoencephalopathy22, metabolic or toxic encephalopathy, and posterior reversible encephalopathy syndrome23.
As minor concern:
- a table showing patients demographic, clinical characteristics, and comorbidities should be added.
>> Thank you for this great suggestion. Added table 2 with the above suggestion.
Reviewer 2 Report
I think that this is a very interesting manuscript in terms of finding MRI features, as there has been a great deal of attention on the examination of covid-related encephalopathy and encephalitis with respect to Covid 19. However, this manuscript has many problems with the methodology and research methods.
(1) There is no comparison between MRIs of Covid 19 patients without encephalopathy and those with Covid 19-associated encephalopathy/encephalitis.
(2) There is no profile of patients with Covid 19-associated encephalopathy/encephalitis.
(3) Is there a difference between MRIs of patients with Covid 19-associated encephalopathy and those with Covid 19-associated encephalitis?
(4) There are quite a number of manuscripts showing that many patients with Covid 19-related encephalopathy/encephalitis will have cerebrovascular disease. What is the difference between MRI of a case of cerebrovascular disease and MRI of a patient with Covid 19-related encephalopathy/encephalitis?
(5) Is there a relationship between clinical symptoms and MRI?
Author Response
(1) There is no comparison between MRIs of Covid 19 patients without encephalopathy and those with Covid-19-associated encephalopathy/encephalitis.
>> Thank you for the comments. Only a fraction of people infected with Covid 19 will develop clinical symptoms of encephalitis. Furthermore, many times these are sedated and intubated, and there is a delay in recognition of underlying encephalitis. We searched these patients from the radiology reports, so it’s not possible to get a precise estimate of the total Covid-19 patients on the inpatient services, and how many of these developed encephalitis. In our study, when we searched the radiology reports, we found 79 patients with encephalitis who underwent MRI, out of which 32 had normal MRIs. 39 patients had abnormal MRI findings.
(2) There is no profile of patients with Covid 19-associated encephalopathy/encephalitis.
>> Thank you for the suggestion. Added table 2.
(3) Is there a difference between MRIs of patients with Covid-19-associated encephalopathy and those with Covid-19-associated encephalitis?
>>> Thank you for the comments. We revisited our criteria and only included patients who meet the criteria for encephalitis (and deleted encephalopathy). Since CSF results are usually negative and tissue diagnosis is not practical, we used the following reference for encephalitis diagnosis.
Venkatesan A, Geocadin RG. Diagnosis and management of acute encephalitis: A practical approach. Neurol Clin Pract 2014;4:206-215.
(4) There are quite several manuscripts showing that many patients with Covid 19-related encephalopathy/encephalitis will have cerebrovascular disease. What is the difference between an MRI of a case of cerebrovascular disease and an MRI of a patient with Covid-19-related encephalopathy/encephalitis?
>>> Thanks for the comment. We didn’t include ischemic infarct patients if there were restricted diffusing lesions on MRI. This was done to avoid overlap between ischemic and encephalitis changes. Regarding microhemorrhages, there is no way MRI can separate if these are due to encephalitis, or ischemic changes. I believe a tissue diagnosis is an option since CSF analysis is usually negative.
(5) Is there a relationship between clinical symptoms and MRI?
>>> thank you, for the comments. Due to the small sample size, we did not find a meaningful relationship between clinical symptoms and MRI.
Reviewer 3 Report
Major:
1. In this article, the authors proposed the criteria for COVID-related encephalitis (Table 1). Even if the criteria were modified from the existing criteria of encephalitis, the attempt to propose new criteria should be mentioned in the Abstract, Introduction (purpose of the study), Results, and Conclusion.
2. The specificity of microbleeds (SWI image findings) to COVID encephalitis was not clear. Could the authors provide clues of linking microbleeds to COVID infection, other than small vessel diseases? If not relevant, the findings of microbleeds might be removed from the typical findings of COVID-infection-related encephalitis.
3. Results: Lacking a diagram of a flow chart of enrollment, which should include information on the number of screenings for eligibility, number of exclusions (the number for each reason of exclusion), and final enrollment.
4. Results: The first paragraph is lacking demographic description of the enrolled patients. Please describe the age, sex, initial clinical symptoms/signs, outcome/mortality, and time from COVID infection to MRI scans. Suggest using a table to summarize the information.
5. Results: Suggest using classification and a table to summarize image findings. For example, list the findings under the headers of MRI sequences, location of lesions, percentage of the findings, and classifications to cortical/subcortical lesions.
6. Results: The current results are relatively weak, especially absence of image-clinical correlations. Suggest adding correlation analysis of the MRI findings/lesion classifications to outcomes (eg., mortality), to severity (eg., GCS coma scale or mechanical ventilation), or to symptoms (eg., seizure, status epilepticus, or psychiatric symptoms).
7. Discussion: The comparisons of the current findings to other types of encephalitis (eg., autoimmune encephalitis or herpetic encephalitis) are absent.
8. Discussion: The common and different findings of this study with previous case reports or case series of COVID-related encephalitis should be mentioned.
Minor:
1. Introduction P2 line47: GBS is a peripheral nervous system disease
2. Introduction P2 line58: The comorbidities seem not relevant to the content in this paragraph.
3. Materials & Methods: The location of image acquisition was not listed.
4. Materials & Methods: The affiliations or locations of the reviewers were not mentioned.
5. Materials & Methods: Did the reviewers use T1-weighted imaging for screening structural anomalies?
6. Materials & Methods: Did not mention using DWI or DWI+ADC for excluding COVID-related ischemic stroke.
Author Response
- In this article, the authors proposed the criteria for COVID-related encephalitis (Table 1). Even if the criteria were modified from the existing criteria of encephalitis, the attempt to propose new criteria should be mentioned in the Abstract, Introduction (purpose of the study), Results, and Conclusion.
>>> Agree! We added that the presence of a positive COVID-19 PCR test is an absolute requirement for inclusion in this study. We explicitly said that it is a Modified Diagnostic Criteria for COVID-19 Encephalitis
- The specificity of microbleeds (SWI image findings) to COVID encephalitis was not clear. Could the authors provide clues of linking microbleeds to COVID infection, other than small vessel diseases? If not relevant, the findings of microbleeds might be removed from the typical findings of COVID-infection-related encephalitis.
>> thanks for the comments. The presence of microbleeds has been reported in multiple prior publications regarding COVID-19.
-Scullen T, Keen J, Mathkour M, et al. Coronavirus 2019 (COVID-19)-Associated Encephalopathies and Cerebrovascular Disease: The New Orleans Experience. World Neurosurg 2020;141:e437-e446
-Radmanesh A, Derman A, Lui YW, et al. COVID-19-associated Diffuse Leukoencephalopathy and Microhemorrhages. Radiology 2020;297:E223-E227
- Sharma R, Nalleballe K, Shah V, et al. Spectrum of Hemorrhagic Encephalitis in COVID-19 Patients: A Case Series and Review. Diagnostics (Basel) 2022;12
However, it is difficult to separate whether the microhemorrhages were the result of tissue ischemia or encephalitis. We decided to include these in the manuscript since it was one of the most commonly observed imaging findings in patients meeting the clinical criteria of encephalitis. Also, we know, other causes of encephalitis, such as herpes encephalitis are also known to demonstrate intracranial microhemorrhages.
- Results: Lacking a diagram of a flow chart of enrollment, which should include information on the number of screenings for eligibility, number of exclusions (the number for each reason of exclusion), and final enrollment.
>> Thank you for the suggestion included in Fig.1
- Results: The first paragraph is lacking demographic description of the enrolled patients. Please describe the age, sex, initial clinical symptoms/signs, outcome/mortality, and time from COVID infection to MRI scans. Suggest using a table to summarize the information.
>> Thank you for the suggestion. Added table 2 with the above information.
- Results: Suggest using classification and a table to summarize image findings. For example, list the findings under the headers of MRI sequences, location of lesions, percentage of the findings, and classifications to cortical/subcortical lesions.
>> Thank you for the suggestion. Added table 3 with the above information.
- Results: The current results are relatively weak, especially absence of image-clinical correlations. Suggest adding correlation analysis of the MRI findings/lesion classifications to outcomes (eg., mortality), to severity (eg., GCS coma scale or mechanical ventilation), or symptoms (eg., seizure, status epilepticus, or psychiatric symptoms).
>> Thank you for the suggestion. Added table 2 with the above information.
- Discussion: The comparisons of the current findings to other types of encephalitis (eg., autoimmune encephalitis or herpetic encephalitis) are absent.
>> Thank you for the suggestion. Added paragraph in the discussion section
While differentiation from other causes of encephalitis can be difficult, other etiologies such as herpes encephalitis present with a fulminant course, with 90% of cases resulting from HSV-1. Diagnosis is established with CSF PCR, although a combination of the clinical scenario, CSF pleocytosis, and elevated protein, and appropriate imaging are usually highly suggestive and permit commencement of treatment 24. Similarly, autoimmune encephalitis can present with the involvement of the mesial temporal lobes and limbic system, manifested by cortical thickening and increased T2/FLAIR signal intensity. However, in approximately 60% of cases, antineuronal antibodies are present, such as the anti-Hu antibody in small cell lung cancer, anti-Ta antibody in testicular cancers, anti-NMDA NR1 in ovarian teratomas, or anti-NMDA NR2 in SLE patients25.
- Discussion: The common and different findings of this study with previous case reports or case series of COVID-related encephalitis should be mentioned.
>> Paragraph 2 in the discussion section.
MRI findings in critically ill or ICU-admitted COVID-19 patients have been previously published by many researchers. The precise pathophysiology is unclear; however, authors have hinted towards immunologic parainfectious processes20, which has been supported by some neuropathologic studies21. Radmanesh et al evoked the assumptions of hypoxia or small-vessel vasculitis16. Direct brain tissue involvement by viral particles has not been endorsed by cerebrospinal fluid analysis, and other hypotheses such as postinfectious demyelinating diseases may be considered. Since most of these patients were admitted to intensive care units for acute respiratory distress syndrome, other assumptions include delayed post-hypoxic leukoencephalopathy22, metabolic or toxic encephalopathy, and posterior reversible encephalopathy syndrome23.
Minor:
- Introduction P2 line47: GBS is a peripheral nervous system disease
>> Thank you. Deleted GBS.
- Introduction P2 line 58: The comorbidities seem not relevant to the content in this paragraph.
>> Agree, deleted.
- Materials & Methods: The location of image acquisition was not listed.
>> Thank you. All MRI images were acquired at 1.5 and 3 T inpatient Philips and Siemens MRI scanners at our tertiary care center affiliated with a medical school.
- Materials & Methods: The affiliations or locations of the reviewers were not mentioned.
>> Thank you. The reviewers are board-certified fellowship trained neuroradiologists working at a tertiary care medical center affiliated with a medical school in the United States.
- Materials & Methods: Did the reviewers use T1-weighted imaging for screening structural anomalies?
>> Yes, we did.
- Materials & Methods: Did not mention using DWI or DWI+ADC for excluding COVID-related ischemic stroke.
> agree, thanks for the comment. We used DWI+ ADC to rule out ischemic stroke
Round 2
Reviewer 2 Report
Thank you for your revise manuscript.
1) The authors should rewrite figure 1. Figure 1 should be redrawn so that the 527 patients are divided into two branches of 456 patients and 71 patients.
2) Please unify whether it is encephalitis or encephalopathy in Table 2 and the text.
3) The sentence in P218 was changed. The signal in the white matter of periventricular region changed around the third ventricular are the other less common findings.
Author Response
1) The authors should rewrite figure 1. Figure 1 should be redrawn so that the 527 patients are divided into two branches of 456 patients and 71 patients.
>> Thank you for the suggestion. Changed figure as suggested.
2) Please unify whether it is encephalitis or encephalopathy in Table 2 and the text.
>> Thank you for the suggestion. Made corrections.
3) The sentence in P218 was changed. The signal in the white matter of periventricular region changed around the third ventricular are the other less common findings.
>> Thank you for the suggestion. Made corrections
Reviewer 3 Report
The first two columns in Table 2 are not informative (GCS and NIHSS) because of missing data in most cases. Suggest delete the two columns.
Author Response
The first two columns in Table 2 are not informative (GCS and NIHSS) because of missing data in most cases. Suggest delete the two columns.
>> Thank you for the suggestion. Deleted those redundant columns.